# The Effects of Ultrasonic Pretreatment and Enzymatic Modification on the Structure, Functional Properties, and In Vitro Digestion of Whey Protein Isolate

**DOI:** 10.3390/foods14091445

**Published:** 2025-04-22

**Authors:** Yingying Yue, Yujun Jiang, Jia Shi

**Affiliations:** 1Department of Food Science, Ministry of Education, Northeast Agricultural University, Harbin 150030, China; yueyingying0406@163.com (Y.Y.); yujun_jiang@163.com (Y.J.); 2Key Laboratory of Infant Formula Food, State Administration for Market Regulation, Harbin 150030, China

**Keywords:** whey protein isolate, dual oxidase system, ultrasound, functional properties, in vitro digestion

## Abstract

In this study, the structure and functional and in vitro digestion properties of whey protein isolate (WPI) modified by ultrasonic pretreatment combined with a double oxidase system containing horseradish peroxidase (HRP), glucose oxidase and D-glucose were assessed. SDS-PAGE results confirmed the occurrence of crosslinking reactions. Ultrasonic treatment significantly increased HRP-mediated WPI crosslinking, as demonstrated by reductions in free amino and sulfhydryl groups. CD and FTIR spectroscopies indicated that the structure of the crosslinked WPI was more stable. The particle size of the modified WPI was significantly reduced, resulting in better colloidal stability. Compared with the untreated WPI, the crosslinked WPI possessed enhanced surface hydrophobicity, gelation properties, emulsion stability, and thermal stability but reduced digestibility. These findings provide new insights into ultrasonication combined with a double oxidase system to further improve the structure and functional properties of proteins and broaden their application range in the food industry.

## 1. Introduction

With the increasing health awareness of consumers, more and more people are beginning to pay attention to protein intake, especially that of high-quality protein. Whey protein isolate (WPI) is obtained by the microfiltration, concentration, and spray drying of whey and contains a high protein content (>90%, *w*/*w*) [1]. In addition, WPI is rich in amino acids and natural whey protein peptides, which have antibacterial, antioxidant, and antihypertensive health functions and are easily digested and absorbed by the human body [2]. These properties make it highly sought after in the fields of nutrition and health foods. WPI, as a component of dairy products, is commonly used as a raw material for food processing because of its superior nutritional value [3]. WPI is also an important component of thickeners, stabilizers, and emulsifiers in food production because of its high physiological activity and functional properties [4]. WPI has excellent solubility because of its external hydrophilic and internal hydrophobic spherical structure. However, the emulsification, gelation, and other functional properties of WPI cannot reach the requirements of the food industry [5]. To achieve these goals of improving the functional properties of WPI, researchers are working to develop methods for modifying WPI. In recent years, many physical (e.g., high-pressure, heating, ultrasonic, and irradiation), chemical (e.g., glycation), and biological enzymatic (e.g., crosslinking and hydrolytic) methods have been extensively studied [6,7,8]. WPI is modified using physical methods with few toxic side effects and at a low cost, but the effects are not obvious. Although chemical modification can effectively improve the functional performance of proteins, some toxic and harmful substances may be produced during or after the reaction, and there are safety problems. Enzymatic modification has the characteristics of mild reaction conditions, strong specificity, safety and reliability, and remarkable modification effects, and it has a wide range of application prospects.

Protein crosslinking is the formation of covalent bonds within or between protein molecules in the presence of chemical reagents or biocatalysts [9,10]. In current research on crosslinking methods, enzyme-mediated crosslinking is a promising approach that can effectively alter proteins’ spatial structures and physicochemical properties. For example, some transferases and oxidoreductases can typically induce protein crosslinking [11]. Horseradish peroxidase (HRP), a heme protein with iron porphyrin as a cofactor, can oxidize tyrosine residues in proteins using H_2_O_2_ produced by glucose-oxidase-catalyzed D-glucose oxidation for protein crosslinking [12]. Jiang et al. [13] have demonstrated that the bi-oxidative enzymatic system promotes the crosslinking of soybean isolate protein. However, the primary constituents of natural whey protein, α-lactalbumin (α-la) and β-lactoglobulin (β-lg), are dense globular proteins, which prevent the reaction between the dual oxidase system and WPI [14]. Thus, it is essential to change the structure of whey proteins to make them easily crosslink with the dual oxidase system. Exposure to enzyme-targeting sites can boost the crosslinking reaction [15], such as by heat treatment [16], superfine grinding treatment [17], and ultrasonication [18]. Ultrasonication is an environmentally friendly technology without exogenous chemical additives, which can quickly destroy the interactions between or within protein molecules via the non-thermal process of mechanical waves and acoustic cavitation, exposing some active sites and promoting enzymatic crosslinking reactions, thereby affecting their physicochemical and functional properties [19]. Li et al. [20] showed that the properties of qingke protein were markedly enhanced following modification with transglutaminase, aided by multiple-frequency ultrasonication, including its foaming, water-holding capacity, and emulsification activity. Similarly, ultrasonic pretreatment improved the characteristics of soy protein isolate gels catalyzed by transglutaminase, such as the strength, storage modulus, consistency, and thermal stability [21]. However, research on the influence of ultrasonication on the structural and functional properties of crosslinked WPI by the dual oxidase system is limited.

Therefore, this study aimed to verify the effects of ultrasonication combined with a dual oxidase system crosslinking treatment on the functional properties of WPI. WPI was pretreated with sonication and then modified with HRP, glucose oxidase, and D-glucose. The SDS-PAGE, secondary structure, particle size, fluorescence spectra, surface hydrophobicity, thermal stability, microstructure, solubility, emulsification, gelation, and digestive properties were evaluated. The results of this study will offer fundamental data for efficient protein modification methods to prepare WPI with better functional features.

## 2. Materials and Methods

### 2.1. Materials

WPI (92.6%, *w*/*w*) was purchased from Hilmar Co., Ltd. (Hilmar, CA, USA). Horseradish peroxidase (HRP, RZ > 2.5, ≥250 U/mg) was bought from Shanghai Yuanye Bio-Technology Co., Ltd. (Shanghai, China). Glucose oxidase (derived from *Aspergillus niger*, >180 U/mg) was obtained from Shanghai Aladdin Biochemical Technology Co., Ltd. (Shanghai, China). Soybean oil was bought from the local supermarket. Sodium dodecyl sulfate (SDS) was obtained from Dalian Meilun Biotechnology Co., Ltd. (Dalian, Liaoning, China). All the other chemical reagents used were analytical grade.

### 2.2. Preparation of Modified WPI

WPI solution, at pH 7 (6% *w*/*v*), was treated with 200 W of ultrasonic power for 5 min and then mixed with HRP, glucose oxidase, and D-glucose at 200 U/g of protein, 6 U/g of protein, and 0.05 mmol/g of protein, respectively. The samples reached a final protein concentration of 5% (*w*/*v*). The mixture was stirred at 37 °C for 3 h, heated at 85 °C for 10 min to inactivate the enzyme, and rapidly cooled to room temperature to obtain modified WPI, denoted as UCWPI. The controls were untreated WPI and individual-enzyme-treated WPI under the same conditions, which were named WPI and CWPI, respectively.

### 2.3. SDS-PAGE Analysis

The occurrence of the crosslinking reaction was confirmed through qualitative analysis by SDS-PAGE. First, 5× loading buffer was added to the protein sample solution (2 g/L), which was boiled for 5 min at 100 °C. Then, electrophoresis was conducted at 80 V. The gel was subjected to staining and destaining, followed by photography using gel-imaging devices.

### 2.4. Determination of Free Amino Group Content

The ortho-phthalaldehyde (OPA) method was used to measure the free amino groups. A mixture of 4 mL of the OPA reagent and 200 μL of the sample solution was allowed to stand for 5 min at room temperature. Measurements were conducted at a wavelength of 340 nm. A standard curve was plotted using L-leucine with a purity of 99%, and the free amino group content was determined according to the regression equation.

### 2.5. Determination of Sulfhydryl (SH) Group Content

The sample (6 mg/mL) was added to a Tris–Glycine buffer containing 8 mol/L of urea. The reaction was conducted for 25 min at 40 °C, and the measured wavelength was 412 nm. The calculation of the SH content was performed using the following designated formula [22]:(1)$$SH\;(\mu mol/g)=\frac{(73.53\times A_{412})\times D}{C}$$
where A_412_ denotes the absorbance measured at 412 nm, D signifies the dilution factor, and C represents the protein concentration being measured (mg/mL).

### 2.6. CD Spectra Analysis

The spectra of the samples were obtained within the wavelength interval from 190 to 260 nm using a circular dichroism spectrometer (Chirascan V100, Applied Photophysics Ltd., Surrey, UK), and their secondary structures were analyzed using CDNN software (version 2.1). The concentration of the samples was 0.1 mg/mL, and the width of the cuvettes was 1 mm.

### 2.7. FTIR Spectroscopy Analysis

FTIR spectroscopy was performed using the KBr-tableting method. The freeze-dried powder sample was mixed with the dried KBr solid in a certain proportion, ground to a uniform and fine powder in an agate mortar, and pressed into a transparent sheet with a tablet press. The sample was placed in the FTIR spectrometer (Nicolet iS50, Thermo Fisher Scientific, Waltham, MA, USA) to scan 32 times in the wavelength range of 4000–400 cm^−1^ at a resolution of 4 cm^−1^.

### 2.8. Fluorescence Spectroscopy Analysis

Emission spectra between 300 and 500 nm were obtained utilizing a fluorescence spectrophotometer (F-7100, Hitachi, Tokyo, Japan) at an excitation wavelength of 280 nm. The slit was 5 nm, and the concentration of the samples was 0.1 mg/mL. The maximum emission wavelengths (λmax) of the samples were recorded. Changes in the tertiary structure of the WPI were assessed based on the change in the fluorescence intensity and the degree of the shift in the maximum emission wavelength.

### 2.9. DSC

The thermal stability was determined via differential scanning calorimetry (DSC250, TA Instruments, New Castle, DE, USA). A sample of 5 mg was tightly sealed in an aluminum dish while using an empty aluminum dish as a control. Nitrogen was used at a flow rate of 50 mL/min as the transfer gas. The samples underwent heating from 30 °C to 180 °C at a rate of 10 °C/min.

### 2.10. Microscopic Morphology Analysis

The lyophilized sample was laid flat on a conductive adhesive, and the operating voltage was set at 5 kV. The morphology was analyzed using scanning electron microscopy (SU 8010, Hitachi, Tokyo, Japan). A sample solution (0.1 mg/mL) was prepared, and a droplet was dispersed on a carbon-coated copper grid, stained with phosphotungstic acid, and then lyophilized to obtain a dry sample. The microstructure was observed using transmission electron microscopy (H-7650, Hitachi, Tokyo, Japan) at 80 kV.

### 2.11. Determination of Particle Size and Zeta Potential

The sample was diluted to 1 mg/mL, and then its particle size and zeta potential were measured using a Malvern Zetasizer Nano system (Malvern Instruments, Ltd., Malvern, UK). Testing was conducted at 25 °C.

### 2.12. Determination of Emulsification Properties

The evaluation of the emulsification capabilities of the samples was conducted following the method described by Jiang et al. [23]. A mixture of soybean oil (3 mL) and the sample solution (9 mL) was homogenized for 2 min at 10,000 rpm using an IKA homogenizer (T18, IKA, Staufen, Germany). Then, 50 μL of the emulsion was removed from the bottom of the container at 0 min and 10 min and diluted with 5 mL of SDS (0.1% *w*/*v*). The measurement wavelength was set at 500 nm. The emulsion stability index (ESI) and emulsification activity index (EAI) were calculated based on the following formulae:(2)$$ESI\;(\%)=\frac{A_{10}}{A_{0}}\times 100$$(3)$$EAI\;(\frac{m^{2}}{g})=\frac{2\times 2.303\times A_{0}\times D}{N\times C\times 10,000}$$
where A_0_ and A_10_ denote the absorbance of the sample at 0 min and 10 min, C signifies the protein concentration prior to the emulsion formation (g/mL), D stands for the dilution factor (D = 100), and N indicates the volumetric fraction occupied by the oil phase (0.25).

### 2.13. Determination of Surface Hydrophobicity

A fluorescent probe method based on 8-phenylamino-1-naphthalenesulfonic acid (ANS) was adopted to analyze the hydrophobic properties of the samples. First, 20 μL of ANS (8 mmol/L) solution was mixed with 4 mL of protein solutions at different concentrations and kept shielded from light for 15 min. The fluorescence intensities of the samples were determined at an excitation wavelength of 390 nm and an emission wavelength of 470 nm. A linear regression analysis was conducted on the fluorescence intensity in relation to the corresponding protein concentration, thus determining the initial linear slope as the surface hydrophobicity index.

### 2.14. Determination of Solubility

The protein samples were centrifuged (at 10,000× *g* for 25 min); then, the supernatant was gathered, and the protein concentration was assessed using the BCA kit method. The solubility was evaluated as the percentage of the protein concentration in the supernatant out of the overall protein concentration [24].

### 2.15. Determination of Gelation Properties

A protein solution at a concentration of 50 mg/mL was mixed with gluconolactone (GDL at 0.2 g/g of protein), heat treated for 15 min at 95 °C, subsequently cooled to room temperature, and stored in a 4 °C freezer overnight. Then, the texture of the gel was analyzed, and the sample to be tested was placed on the assay platform with a probe (model p/0.5S). The gels were compressed to 50% of their original height at a constant speed of 1 mm/s. The trigger force was 10 g.

### 2.16. In Vitro Digestion Analysis

Different protein solutions were taken, and an equal volume of gastric juice (SGF, which contained 2000 U/mL of pepsin) was added and mixed well. The reaction occurred for 2 h at 37 °C, at which point, the pH of the solution was 3. Subsequently, the gastric digest was mixed with an equal volume of intestinal fluid (SIF, which contained 100 U/mL of trypsin and 10 mmol/L of bile salts). The reaction for the digestion took place at 37 °C for 2 h, at which point, the pH of the solution was 7. After the digestion reaction was completed, the solution was boiled for 5 min then centrifuged at 8000 rpm at 4 °C for 15 min [25], and the supernatant was taken for further analysis. The degree of the hydrolysis was assessed based on the protein and free amino group contents, and the digested products were characterized using SDS-PAGE and particle size and zeta potential measurements.

### 2.17. Statistical Analysis

All the trials were carried out in three replications. The final data were expressed as means ± standard deviations (SDs). A significant difference (*p* < 0.05) analysis was conducted utilizing IBM SPSS Statistics 27 and one-way ANOVA.

## 3. Results and Discussion

### 3.1. SDS-PAGE

The occurrence of protein crosslinking catalyzed with the double oxidase system was verified by SDS-PAGE, which was manifested by the change in the WPI’s molecular weight. As shown in Figure 1A, two major bands, corresponding to α-la and β-lg monomers, were seen in the WPI lane (Lane 1), located at about 14 kDa and 17 kDa, respectively. Compared with WPI, CWPI (Lane 2) and UCWPI (Lane 3) showed high-molecular-weight polymers, and the corresponding α-la and β-lg contents decreased, indicating that the double oxidase system could catalyze WPI crosslinking. The presence of macromolecules in the electropherogram of the caseinate has been reported in the presence of dual oxidase [26]. After sonication, the corresponding α-la and β-lg contents did not change significantly, indicating that ultrasonic pretreatment had little effect on the crosslinked WPI’s molecular weight in the double oxidase system. These results indicated that the dual oxidase system induced the crosslinking of the WPI.

### 3.2. Free Amino Group Content

As illustrated in Figure 1B, the enzymatic crosslinking alone did not significantly affect the free amino group content (*p* > 0.05). In contrast, when the WPI was treated with ultrasonication and enzymatic crosslinking, the free amino group content was significantly reduced by 9.33%. The reduction in the content of the free amino groups explains the level of crosslinking in the WPI by the double oxidase system to some extent because HRP is an oxidoreductase enzyme that oxidizes tyrosine residues in proteins in the presence of H_2_O_2_ to form dityrosine residues, resulting in covalent crosslinking within or between proteins. Sonication can destroy the molecular structure of the protein, providing more active regions [27]. When the WPI was further modified with enzymes, the interaction between the substrate and the enzyme was enhanced, leading to a decrease in the free amino group content within the protein. These findings indicated that ultrasonic pretreatment could enhance the extent of the crosslinking reaction catalyzed by the double oxidase system.

### 3.3. SH Group Content

The measurement of the sulfhydryl group contents is based on the principle that proteins with sulfhydryl groups readily react with Ellman’s reagent. Figure 1C illustrates that the sulfhydryl group contents of both the CWPI and UCWPI were lower than that of the WPI (*p* < 0.05), and the sulfhydryl group content of the CWPI decreased from 11.35 ± 0.12 nmol/mg of protein to 5.18 ± 0.13 nmol/mg of protein, a decrease of 54.36%; the sulfhydryl group content of the UCWPI decreased from 11.35 ± 0.12 nmol/mg of protein to 4.76 ± 0.17 nmol/mg of protein, a decrease of 58.06%. In general, under the condition of heating or oxidation, the free sulfhydryl group in proteins is easily converted to disulfide bonds, which are subsequently crosslinked to create high-molecular-weight polymers [28]. SDS-PAGE results also confirmed this phenomenon. During the preparation of CWPI and UCWPI, glucose oxidase oxidizes D-glucose to produce H_2_O_2_. Then, it creates an oxidative environment, which promotes the oxidation of free sulfhydryl groups. As a result, CWPI and UCWPI had lower levels of sulfhydryl groups. The TGase-catalyzed formation of soybean isolate protein gels has been reported to promote disulfide bond formation while decreasing the sulfhydryl group content [21]. In addition, Li et al. [29] showed that sulfhydryl groups may form disulfide bonds through oxidative reactions following the TGase crosslinking of the WPI. This process led to a reduction of sulfhydryl groups. Compared to the CWPI, the UCWPI had a lower sulfhydryl group content (*p* < 0.05) because ultrasonication promoted the enzymatic crosslinking of the WPI, resulting in a decrease in sulfhydryl groups.

### 3.4. CD Spectra

CD spectroscopy has been used to extensively evaluate the secondary structures of proteins, as various band positions and peak intensities in CD spectra vary among different proteins. The CD spectra of the WPI, CWPI, and UCWPI are shown in Figure 2A. The outcomes are shown in Figure 2B. The secondary structure of the WPI consisted of 10.10% α-helixes, 39.03% β-sheets, 20.19% β-turns, and 30.68% random coils. After the double-oxidase-induced crosslinking, the secondary structure of the WPI changed, with the α-helix content increasing by 29.70% and the β-sheet content decreasing by 15.83%. After ultrasonication combined with the enzymatic treatment, the α-helix content increased by 34.06%, and the β-sheet content decreased by 16.06%. Compared with the WPI, both the CWPI and UCWPI showed increased percentages of α-helix and reduced percentages of β-sheet contents, suggesting that the secondary structure of the WPI was more stable after modification [30]. The UCWPI was more crosslinked than the CWPI, so it had the highest α-helix content and the lowest β-sheet content. For both the CWPI and UCWPI, two tyrosine residues were covalently linked to form a dityrosine residue. As a result, the modified WPI by double oxidase system produced a stable secondary structure.

### 3.5. FTIR Spectroscopy

In general, peak shape changes and position shifts in FTIR spectra can indicate changes in the secondary structures of proteins. The amide A band, amide I band, and amide II band are the three major characteristic peaks in the WPI spectrum, corresponding to 3500–3000 cm^−1^, 1700–1600 cm^−1^, and 1600–1500 cm^−1^, respectively. According to Figure 3A, as an amphiphilic protein, WPI possessed a spectrum that exhibited sharp peaks at 3289.28 cm^−1^ and 2961.94 cm^−1^, pertaining to the tensile oscillations of hydrophilic N-H and O-H and hydrophobic C-H, respectively. In addition, characteristic absorption peaks of amide bonds were observed at 1647.51 cm^−1^ and 1543.12 cm^−1^, corresponding to C-O stretching and C-N and N-H stretching, respectively. The spectrum of the enzyme-modified crosslinked WPI showed a blueshift in the most intense absorption peak between the amide A bands and a decrease in the amplitude of that peak compared to the counterpart absorption peak for the WPI, suggesting that the N-H stretching of the amide A bands changed and that the -NH_2_ group in the protein was involved in the reaction and indicating that there was an enhanced interaction between the protein molecules, mainly through hydrogen bonding [31]. The peaks in the CWPI and UCWPI spectra both had higher wavenumbers than that in the WPI spectrum, corresponding to the peak of the amide I band, increasing from 1647.51 cm^−1^ to 1654.82 cm^−1^. These increases in wavenumbers indicated a change in the C-O bonding in the protein molecules [15]. The change in the strength or position of the functional groups in the protein indicated a change in the secondary structure of the modified WPI. The secondary structure of the UCWPI was affected by the ultrasonic pretreatment in addition to the enzyme.

### 3.6. Fluorescence Spectroscopy

The amino acids with fluorescence properties in protein molecules are tryptophan, tyrosine, and phenylalanine (Trp, Tyr, and Phe), and endogenous fluorescence primarily arises from Trp and Tyr residues. These aromatic amino acid residues in WPI are selectively excited, and the alterations in the tertiary structure of the WPI are assessed according to the fluorescence intensity [32]. Figure 3B shows the changes in the intensity and the maximum emission wavelength (λmax) of the endogenous fluorescence spectra of the WPI before and after the treatment, with emission wavelengths in the 300–500 nm range. The fluorescence intensity of the WPI was enhanced by the dual oxidase systemic crosslinking, suggesting that some fluorescent amino acid residues were favorably exposed by the enzyme-mediated crosslinking. Sonication enhanced the fluorescence intensity of the enzymatically crosslinked WPI. Sonication exposed more active regions sensitive to the dual oxidase system and increased the degree of crosslinking, resulting in more fluorescent amino acid residues being exposed. In addition, the λmax values of the WPI and crosslinked WPI were 358 and 362 nm, respectively, which were slightly redshifted, suggesting that the structure of the WPI changed after crosslinking by the dual oxidase system [33].

### 3.7. Thermal Stability

The thermal denaturation temperature (Tp) and enthalpy change (ΔH) can be used as indicators of protein denaturation, with Tp indicating the thermal stability of the protein and ΔH being inversely related to the extent of the protein denaturation. Higher values of Tp and lower values of ΔH indicate increased thermal stability of proteins [34]. The DSC thermal analysis spectra for the WPI, CWPI, and UCWPI are shown in Figure 4A. The Tp values were 149.95 °C, 150.57 °C, and 154.47 °C, respectively, and the ΔH values were 26.07 J/g, 3.33 J/g, and 3.29 J/g, respectively (Figure 4B). The Tp value of the modified product increased, and the ΔH value decreased. This suggested that the thermal stability of the WPI improved following enzymatic crosslinking, which can be attributed to the formation of covalent bonds as a result of the enzymatic treatment. The results of Li et al. [35] showed similar trends. The ultrasonic pretreatment led to an increase in the Tp value of the WPI and a reduction in ΔH, suggesting that the ultrasonic pretreatment improved the thermal stability of the WPI. The rise in Tp can be ascribed to the ultrasonic pretreatment altering the WPI structure and promoting the crosslinking reaction, which resulted in the formation of an ordered and stable structure.

### 3.8. Microscopic Morphology

The WPI, CWPI, and UCWPI morphologies were observed using scanning electron microscopy at different magnifications (500× and 1000×) and transmission electron microscopy. Compared with the untreated WPI, the interaction within the protein molecule of CWPI and UCWPI was enhanced, and the structures became more compact after the crosslinking of the double oxidase system (Figure 5A). Fan et al. [36] reported tyrosinase- and laccase-induced protein crosslinking, and their SEM images also showed increasingly continuous and compact protein networks. The transmission electron microscopy images show the particle shapes and sizes of the samples. The WPI, CWPI, and UCWPI particles all exhibited spherical shapes, and the UCWPI particles were smaller in size than the WPI and CWPI particles, indicating that the modification of the WPI by ultrasonic pretreatment and the double oxidase system led to a decrease in the particle size (Figure 5B). This was consistent with the measurement of particle size.

### 3.9. Particle Size and Zeta Potential Measurements

Figure 6A shows that the average particle sizes of the WPI, CWPI, and UCWPI were 378.97 ± 5.82 nm, 176.37 ± 3.26 nm, and 163.63 ± 5.11 nm, respectively. Compared with the WPI, the CWPI and UCWPI had smaller particle sizes (*p* < 0.05). This suggests that crosslinked proteins can form smaller colloidal particles in solution, which results in better colloidal stability of the solution, as smaller-sized protein aggregates should have greater resistance to aggregation or coagulation. Han et al. [12] measured the hydrodynamic radius of gelatin in dispersions by DLS and observed that crosslinked gelatin formed smaller aggregates than bovine gelatin. This finding suggested that crosslinked gelatin might produce smaller aggregates while maintaining good colloidal stability.

The zeta potential values of the WPI, CWPI, and UCWPI were −31.10 ± 1.28 mV, −31.90 ± 3.08 mV, and −38.10 ± 2.72 mV, respectively (Figure 6B). The absolute values of the zeta potentials of all the samples exceeded 30 mV, indicating that the samples were stable [37]. The CWPI and UCWPI had greater absolute values of the zeta potentials than the WPI, and the results again demonstrated that crosslinked proteins could provide better colloidal protection against the aggregation of colloidal particles than untreated proteins. Li et al. [3] showed that the absolute values of the zeta potentials of TGase-modified WPI were all significantly higher than those of the untreated WPI. That is, crosslinked proteins have excellent colloidal stability in solution.

### 3.10. Emulsification Properties

The evaluation of the emulsification properties of proteins involves the emulsification activity and emulsion stability. The EAI for the WPI affected by the dual oxidase system was markedly reduced (*p* < 0.05) compared to that for the untreated WPI (Figure 7A), and the emulsification activities of the CWPI and UCWPI were reduced by 57.00% and 66.69%, respectively, compared with that of the WPI. This is because enzymatic treatment results in crosslinking between WPI molecules, forming more macromolecules. Large particles of proteins cannot maintain flexibility when adsorbed onto the surfaces of oil droplets, which reduces the interfacial adsorption capacity of the proteins, thus leading to their lower EAIs [38].

The emulsion stabilities of the CWPI and UCWPI were significantly improved compared to that of the WPI (*p* < 0.05) (Figure 7B). The double oxidase system crosslinked the WPI to produce stable macromolecules that formed a more rigid interfacial film with an increase in the viscosity of the proteins, which decreased the oil spreading of the emulsion, resulting in an increase in the protein’s ESI. In addition, this can be explained by the fact that the increased molecular weight of the polypeptide chains may result in diminished flexibility and the reduced ability of the protein to unfold at the oil–water interface [17], thus increasing the emulsion stability.

### 3.11. Surface Hydrophobicity

The surface hydrophobicity of a protein is contingent upon the exposure of hydrophobic groups, and the more hydrophobic residues that are exposed to the surface of the molecule, the stronger the surface hydrophobicity is [39]. The exogenous fluorescent substance ANS is a hydrophobic dye that predominantly binds to the exposed hydrophobic sites on the protein surface, significantly increasing the fluorescence intensity of the hydrophobic sites. Enzyme-catalyzed protein crosslinking can promote the stretching of the protein’s spatial structure [40]. Figure 7C illustrates that the surface hydrophobicity of the WPI crosslinked with the dual oxidase system was markedly increased (*p* < 0.05). This suggests that the crosslinking process may cause the polypeptide chain of the WPI to unfold, thereby exposing the hydrophobic amino acid residues. Ultrasonication is able to significantly enhance the hydrophobic characteristics of the crosslinked WPI (*p* < 0.05). This enhancement may result from the generation of strong shock waves, shear forces, and turbulence by ultrasonication, which exposes additional active sites on the WPI surface and increases the crosslinking extent, thereby exposing more hydrophobic amino acid residues.

### 3.12. Solubility

The solubilities of the WPI, CWPI, and UCWPI were 68.95 ± 0.92%, 67.63 ± 2.76%, and 64.95 ± 0.45%, respectively (Figure 7D), and there was no significant effect of the crosslinking reaction, mediated by the dual-oxidase system, on the solubility of the WPI (*p* > 0.05). Therefore, the crosslinking treatment with the dual oxidase system hardly affects the solubility of the WPI.

### 3.13. Gelation Properties

As shown in Figure 8A,B, compared with the WPI, the CWPI and UCWPI possessed improved gel hardness and gel gumminess properties, with UCWPI possessing the maximum properties. This was probably because ultrasonication promoted the intermolecular or intramolecular crosslinking of the WPI, and the gel structure was relatively dense, which improved the hardness and gumminess of the gel [41]. Cui et al. [42] also reported that sonication pretreatment increased the hardness and gumminess of enzymatically crosslinked WPI gels.

### 3.14. In Vitro Digestion

As can be seen from Figure 9, the degrees of hydrolysis of the CWPI and UCWPI were lower than that of the WPI. The crosslinked proteins generally exhibited lower digestibility compared to that of the native proteins [43]. This can be understood as the modification of the WPI by the dual oxidase system, causing crosslinks within and between the protein molecules and forming new bonds that trypsin cannot recogniz, which reduces the digestion effect. After gastric digestion, the characteristic bands of the protein became shallow. The protein molecules mainly diffused to below the molecular weight of 15 kDa (Figure 1A). After intestinal digestion, the characteristic bands of the protein disappeared, and the protein molecules became small peptides (Figure 1A), indicating that the structure and molecular weight of the protein changed significantly after digestion. A study by Glusac et al. [44] showed similar results. The particle size and zeta potential of the protein sample changed significantly after digestion. The increases in the particle sizes for the three proteins were notable (Figure 6A), as the hydrophobic groups were exposed, leading to molecular aggregation because of the action of the trypsin [45]. The absolute value of the zeta potential of the digestion product was elevated (Figure 6B), which was attributed to the action of bile salts [46]. The sonication pretreatment and crosslinking treatment with the dual oxidase system could increase the particle size and the absolute value of the zeta potential of the WPI after digestion.

## 4. Conclusions

In this work, we systematically analyzed the effects of different treatments on WPI’s structure, functional properties, and digestion properties using ultrasonic pretreatment combined with a double oxidase system containing HRP, glucose oxidase and D-glucose crosslinked with the WPI. SDS-PAGE analysis showed that high-molecular-weight polymers were formed after enzymatic crosslinking the WPI. Compared with the untreated WPI, the modified WPI contained decreased contents of free amino and sulfhydryl groups. In addition, the modified WPI’s particle size was smaller and secondary structure was more stable than those of the untreated WPI. Synergistic ultrasonic pretreatment and enzymatic crosslinking significantly improved the surface hydrophobicity, thermal stability, emulsion stability, and gelation properties of the WPI. In vitro digestion experiments showed that the digestibility of the modified WPI was reduced. This study demonstrated that ultrasonic pretreatment combined with the double oxidase system could effectively increase the degree of crosslinking of the WPI and HRP and improve the functional properties of the WPI. Therefore, ultrasonication is an effective technique to modify WPI to improve HRP-mediated crosslinking reactions, thereby broadening the application of WPI in the food industry.

## Figures and Tables

**Figure 1 foods-14-01445-f001:**
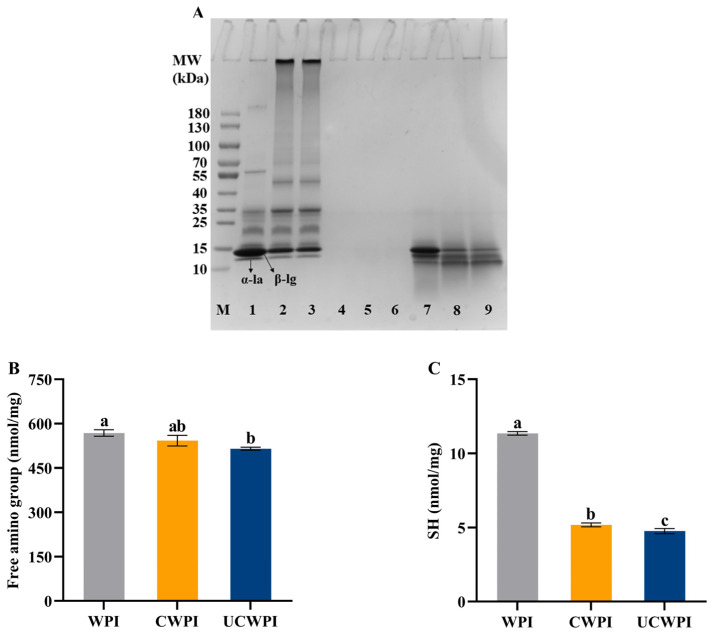
SDS-PAGE (**A**) of WPI, CWPI, and UCWPI before and after digestion. (Lane M: marker; Lane 1: WPI; Lane 2: CWPI; Lane 3: UCWPI; Lanes 4, 5, and 6: WPI, CWPI, and UCWPI intestinal digests, respectively; Lanes 7, 8, and 9: WPI, CWPI, and UCWP gastric digests, respectively). Free amino group (**B**) and SH (**C**) of WPI, CWPI, and UCWPI. Different lowercase letters indicate statistically significant differences (*p* < 0.05).

**Figure 2 foods-14-01445-f002:**
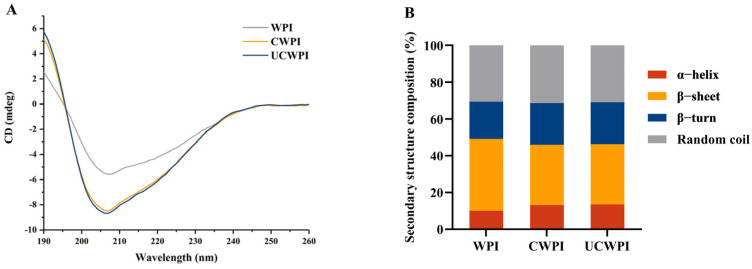
CD spectra (**A**) and secondary structural analysis (**B**) of the WPI, CWPI, and UCWPI.

**Figure 3 foods-14-01445-f003:**
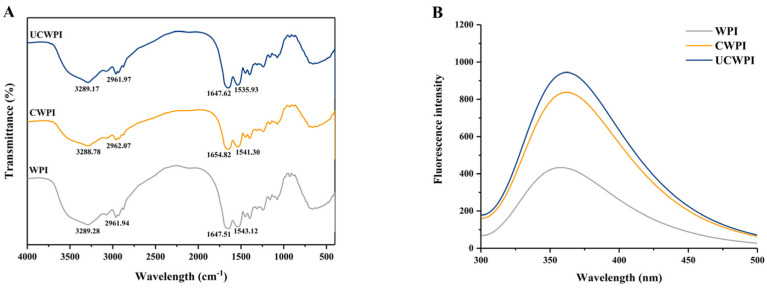
FTIR (**A**) and fluorescence spectra (**B**) of the WPI, CWPI, and UCWPI.

**Figure 4 foods-14-01445-f004:**
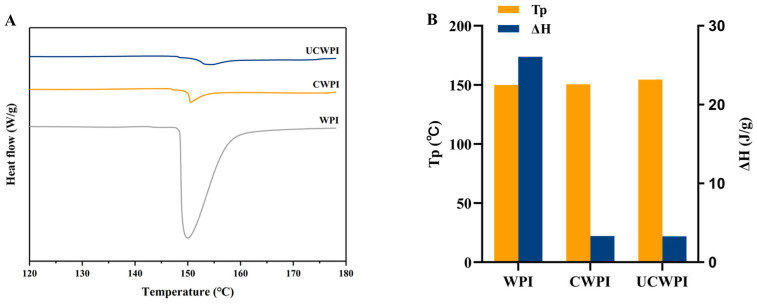
DSC spectra (**A**) and thermal stability analysis (**B**) of the WPI, CWPI, and UCWPI.

**Figure 5 foods-14-01445-f005:**
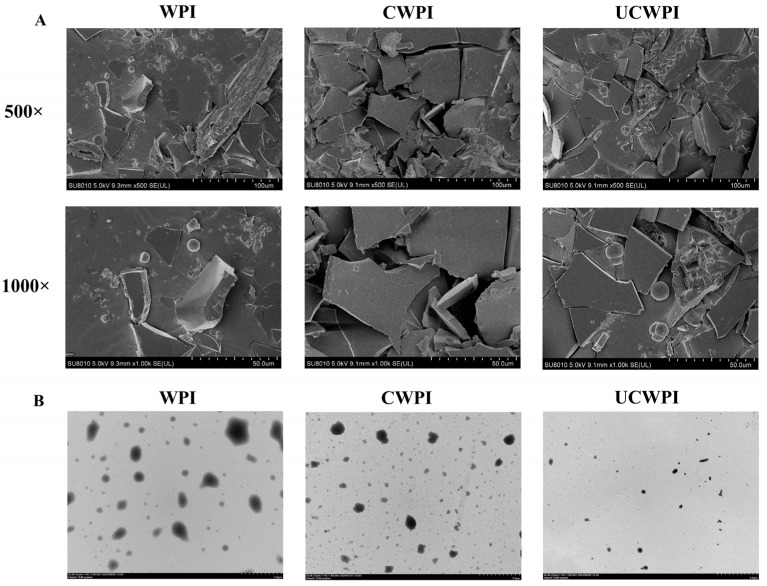
SEM images (**A**) of the WPI, CWPI, and UCWPI at 500× and 1000× magnifications. TEM images (**B**) of the WPI, CWPI, and UCWPI.

**Figure 6 foods-14-01445-f006:**
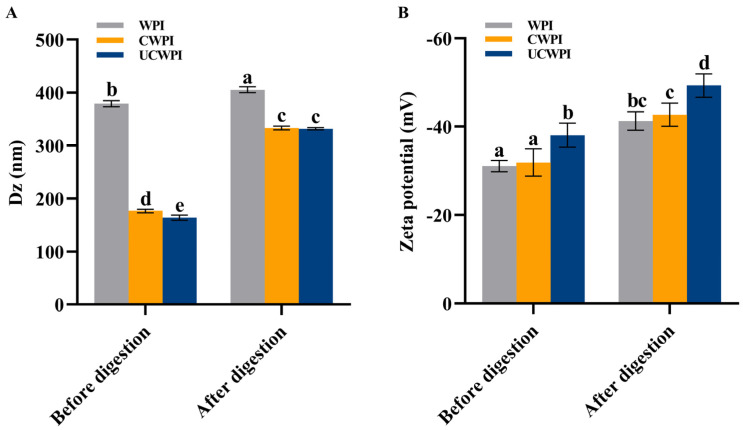
Particle size (**A**) and zeta potential (**B**) of the WPI, CWPI, and UCWPI before and after digestion. Different lowercase letters indicate statistically significant differences (*p* < 0.05).

**Figure 7 foods-14-01445-f007:**
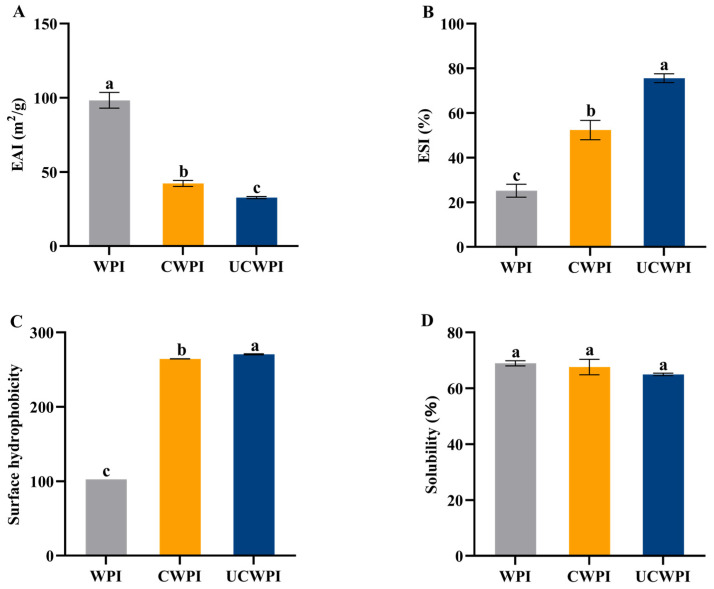
Emulsification activities (**A**), emulsion stabilities (**B**), surface hydrophobicities (**C**), and solubilities (**D**) of the WPI, CWPI, and UCWPI. Different lowercase letters indicate statistically significant differences (*p* < 0.05).

**Figure 8 foods-14-01445-f008:**
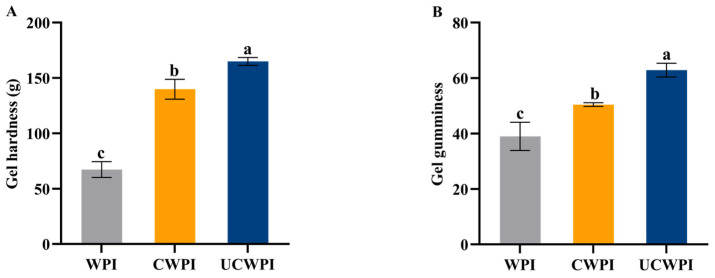
The gel hardness (**A**) and gel gumminess (**B**) of the WPI, CWPI, and UCWPI. Different lowercase letters indicate statistically significant differences (*p* < 0.05).

**Figure 9 foods-14-01445-f009:**
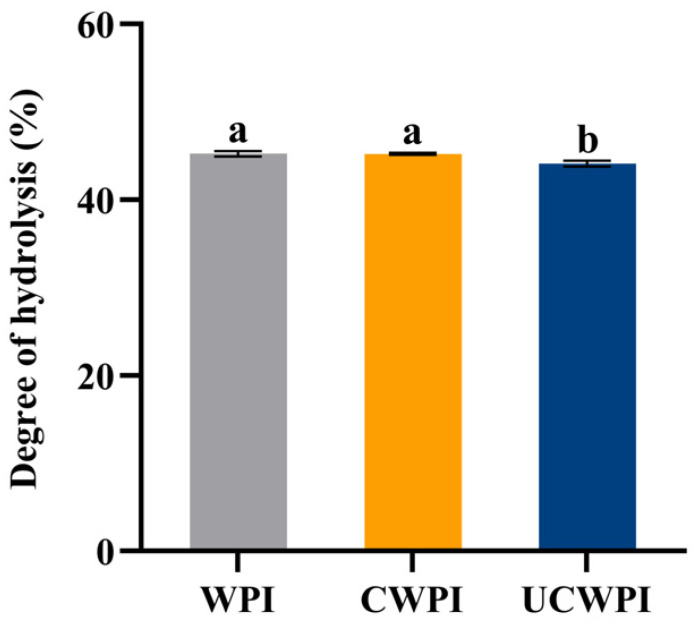
Degrees of hydrolysis of the WPI, CWPI, and UCWPI digestion products. Different lowercase letters indicate statistically significant differences (*p* < 0.05).

## Data Availability

The original contributions presented in the study are included in the article. Further inquiries can be directed to the corresponding author.

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
