# Peer review of "The Effects of Ultrasonic Pretreatment and Enzymatic Modification on the Structure, Functional Properties, and In Vitro Digestion of Whey Protein Isolate"

_foods, 2025, doi:10.3390/foods14091445_

Round 1
Reviewer 1 Report
Comments and Suggestions for Authors
Dear authors,
After reviewing your manuscript, I would like to make the following suggestions: comments.
General: The article presents an interesting idea about the combined effect of ultrasound with dual enzymatic modification of WPI. The content is comprehensive, and interesting results regarding the proposed methodology are presented.
My most significant concern is that, although the idea is exciting, the methodological approach is not clarified in a way that allows for individual analysis to be seen to understand the overall behavior of the treatments. I am referring specifically to the lack of clarity regarding how the individual treatments are structured. For example, there is no mention of ultrasound + HRP or ultrasound + D-glucose oxidase, double oxidase, HRP, or D-glucose oxidase. It only mentions that individual treatments were performed but does not mention which ones and why.
Additionally, the results do not appear conclusive in demonstrating that ultrasound was a major cross-linking promoter. While this is not a problem, more discussion is needed about why a more noticeable effect was not seen.
Specifics:
Introduction
The introduction is concise and lacks information that helps understand the impact of the research; for example, what is the importance of WPI at the market level? Additionally, it fails to mention the advantages of enzymatic cross-linking of WPI compared to physical and chemical methods. I think this would greatly aid the manuscript.
Methodology:
The methodology is too concise, and many details essential for the proper reproducibility of the results are missing. For example, from sections 2.6 to 2.11, the equipment used is not even mentioned, and some important parameters are missing.
L85. How was the WPI solution prepared? This is vitally important because everything else depends on this dispersion. Was a buffer used to control the pH, for example?
L85. What were the characteristics of the ultrasound equipment? How was the temperature controlled to prevent it from varying the results obtained?
L92. Why was a proximal chemical analysis of the raw material not considered?
L117. Why is there no mention of what type of static comparison was performed? The results present these, but it does not mention how they were performed.
Results
Overall, the section needs work to describe the results adequately. For example, monomer positions are mentioned, but no link is made to which MW should be present. It only mentions their presence or whether they decreased, but no further information is given. The information provided in the figure caption should be in the main text of the section.
L185-186. The image is confusing; graphic elements should be placed in Lane 1 to make it clearer.
L190. It is mentioned that After sonication, the corresponding α-Ia and β-Ig contents were lower; however, when looking at the F1a image, the bands present appear the same.
L239-241. It is mentioned that: -the helix increased by 29.70%, and the β-sheet content decreased by 15.83%. After ultrasonic treatment combined with enzyme treatment, the α-helix in240 increased by 34.06%, and the β-sheet content decreased by 16.06%. However, with a difference of 4 °C, can we conclude that cross-linking is indeed present? Why don't the data show deviations? Furthermore, there are no standard deviations.
Author Response
Reviewer #1:
After reviewing your manuscript, I would like to make the following suggestions: comments.
Reply: Thank you for your valuable comments. We have done our best effort to revise the manuscript to ensure its quality.
General: The article presents an interesting idea about the combined effect of ultrasound with dual enzymatic modification of WPI. The content is comprehensive, and interesting results regarding the proposed methodology are presented.
My most significant concern is that, although the idea is exciting, the methodological approach is not clarified in a way that allows for individual analysis to be seen to understand the overall behavior of the treatments. I am referring specifically to the lack of clarity regarding how the individual treatments are structured. For example, there is no mention of ultrasound + HRP or ultrasound + D-glucose oxidase, double oxidase, HRP, or D-glucose oxidase. It only mentions that individual treatments were performed but does not mention which ones and why.
Reply: Thanks for your comments.
Previous studies have used HRP, D-glucose oxidase and double oxidase system to cross-link proteins respectively, and the results showed that the double oxidase system catalyzes protein cross-linking to the highest degree, so we directly selected the double oxidase system to crosslink whey protein isolate, and did not set up a separate HRP and D-glucose oxidase treatment group.
Chang C H, Zhao X H. In vitro digestibility and rheological properties of caseinates treated by an oxidative system containing horseradish peroxidase, glucose oxidase and glucose[J]. International Dairy Journal, 2012, 27(1-2): 47-52. https://doi.org/10.1016/j.idairyj.2012.07.004
The purpose of ultrasound in this study was to pretreat the protein so that the structure of the protein was opened, more enzyme action points were exposed, and the reaction of the protein with the double oxidase system was promoted. Therefore, ultrasound + HRP or ultrasound + D-glucose oxidase treatments were not studied separately.
Thanks again!
Additionally, the results do not appear conclusive in demonstrating that ultrasound was a major cross-linking promoter. While this is not a problem, more discussion is needed about why a more noticeable effect was not seen.
Reply: Thank you for your suggestion.
According to the significance analysis of the indexes measured in this study, ultrasound promoted the cross-linking reaction between whey protein isolate and the double oxidase system but had no significant effect on the solubility of protein. In addition, the change in molecular weight on the SDS-PAGE electrophoresis band was not very significant. Based on your valuable suggestions, we revised the results and analysis of some indicators to make them clearer. Please see revisions to lines 216-219, 257-258, and 341 of the revised manuscript.
Thanks again!
Specifics:
Introduction
The introduction is concise and lacks information that helps understand the impact of the research; for example, what is the importance of WPI at the market level? Additionally, it fails to mention the advantages of enzymatic cross-linking of WPI compared to physical and chemical methods. I think this would greatly aid the manuscript.
Reply: Thank you for your valuable suggestion.
We revised the introduction based on your valuable suggestions. Please refer to the revisions to lines 29-35 and 44-52 of the revised manuscript.
Methodology:
The methodology is too concise, and many details essential for the proper reproducibility of the results are missing. For example, from sections 2.6 to 2.11, the equipment used is not even mentioned, and some important parameters are missing.
Reply: Thanks for your comments.
We revised the method section based on your valuable suggestions. Please refer to the revisions in sections 2.6 to 2.11 of the revised manuscript.
L85. How was the WPI solution prepared? This is vitally important because everything else depends on this dispersion. Was a buffer used to control the pH, for example?
Reply: Thanks for your comments.
Weighed an appropriate amount of WPI powder, dissolved it in deionized water to make the concentration of the solution 6%, fully dissolved, and adjusted the pH of the solution to 7 with 0.2M NaOH.
L85. What were the characteristics of the ultrasound equipment? How was the temperature controlled to prevent it from varying the results obtained?
Reply: Thanks for your comments.
Ultrasound equipment is characterized by the ability to adjust power and time, and the temperature can be displayed. During sonication, the sample was immersed in an ice water bath to remove the heat generated by the sonication treatment, and the temperature was controlled in the range of 25 ± 2 ℃.
L92. Why was a proximal chemical analysis of the raw material not considered?
Reply: Thanks for your comments.
Whey protein isolate was provided by Hilmar Co., Ltd. (CA, USA), which was previously determined by the Kjeldahl nitrogen method with a relatively high purity of 92.6%. The horseradish peroxidase and glucose oxidase used were commercially available. Commercially available products have been quality-validated, so we did not perform additional proximal chemical analysis. Horseradish peroxidase (HRP, RZ>2.5, ≥250 U/mg) was bought from Shanghai Yuanye Bio-Technology Co., Ltd. (Shanghai, China). Glucose oxidase (from Aspergillus niger, >180 U/mg) was obtained from Shanghai Aladdin Biochemical Technology Co., Ltd. (Shanghai, China).
L117. Why is there no mention of what type of static comparison was performed? The results present these, but it does not mention how they were performed.
Reply: Thanks for your comments.
In this study, the changes in the tertiary structure of WPI were evaluated by comparing the changes in fluorescence intensity and maximum emission wavelength (λmax) of the samples before and after treatment by measuring fluorescence spectra. We refined the methodology for this section. Please refer to the revisions in Section 2.8 of the revised manuscript.
Results
Overall, the section needs work to describe the results adequately. For example, monomer positions are mentioned, but no link is made to which MW should be present. It only mentions their presence or whether they decreased, but no further information is given. The information provided in the figure caption should be in the main text of the section.
Reply: Thank you for your valuable suggestion.
Based on your valuable suggestions, we revised the analysis of the results in this section to include the information provided in the title of the figure in the main text. Please refer to the revisions in lines 212-213 of the revised manuscript.
L185-186. The image is confusing; graphic elements should be placed in Lane 1 to make it clearer.
Reply: Thank you for your suggestion.
We have placed the graphic elements in Lane 1. Please see Figure 1A in the revised manuscript.
L190. It is mentioned that After sonication, the corresponding α-Ia and β-Ig contents were lower; however, when looking at the F1a image, the bands present appear the same.
Reply: Thanks for your comments.
After ultrasound combined with enzyme cross-linking, the electrophoresis bands did not change significantly, but if you zoom in on the image, you can see some differences. We revised the analysis in that section. Please refer to the revisions of lines 216-219 in the revised manuscript.
Thanks again!
L239-241. It is mentioned that: -the helix increased by 29.70%, and the β-sheet content decreased by 15.83%. After ultrasonic treatment combined with enzyme treatment, the α-helix in240 increased by 34.06%, and the β-sheet content decreased by 16.06%. However, with a difference of 4 °C, can we conclude that cross-linking is indeed present? Why don't the data show deviations? Furthermore, there are no standard deviations.
Reply: Thanks for your comments.
Based on the measurement results of free amino and free sulfhydryl groups, it was indeed proved that ultrasound promoted the cross-linking of whey protein isolate by the double oxidase system. Therefore, it was speculated that the increased α-helix content and decreased β-sheet content of whey protein isolate after ultrasound combined with double oxidase system treatment were caused by cross-linking.
The data did not have a standard deviation because some of the previous literature also did not show a standard deviation for the protein secondary structure content, some of which we have listed below.
(1) Guo S, Guo Q, Zhang Y, et al. Preparation of enzymatically cross-linked α-lactalbumin nanoparticles and their application for encapsulating lycopene[J]. Food Chemistry, 2023, 429: 136394. https://doi.org/10.1016/j.foodchem.2023.136394
(2) Pu P, Zheng X, Jiao L, et al. Six flavonoids inhibit the antigenicity of β-lactoglobulin by noncovalent interactions: A spectroscopic and molecular docking study[J]. Food chemistry, 2021, 339: 128106. https://doi.org/10.1016/j.foodchem.2020.128106
(3) Liu X, Song Q, Li X, et al. Effects of different dietary polyphenols on conformational changes and functional properties of protein–polyphenol covalent complexes[J]. Food Chemistry, 2021, 361: 130071. https://doi.org/10.1016/j.foodchem.2021.130071
(4) Li Y, Su J, Cavaco-Paulo A. Laccase-catalyzed cross-linking of BSA mediated by tyrosine[J]. International Journal of Biological Macromolecules, 2021, 166: 798-805. https://doi.org/10.1016/j.ijbiomac.2020.10.237
(5) Liu K, Chen S, Chen H, et al. Cross-linked ovalbumin catalyzed by polyphenol oxidase: Preparation, structure and potential allergenicity[J]. International Journal of Biological Macromolecules, 2018, 107: 2057-2064. https://doi.org/10.1016/j.ijbiomac.2017.10.072
(6) Wang N, Ma Z, Ma L, et al. Synergistic modification of structural and functional characteristics of whey protein isolate by soybean isoflavones non-covalent binding and succinylation treatment: A focus on emulsion stability[J]. Food Hydrocolloids, 2023, 144: 108994. https://doi.org/10.1016/j.foodhyd.2023.108994
(7) Li X, Bai H, Wu Y, et al. Structural analysis and allergenicity assessment of an enzymatically cross-linked bovine α-lactalbumin polymer[J]. Food & function, 2020, 11(1): 628-639. https://doi.org/10.1039/c9fo02238d
Reviewer 2 Report
Comments and Suggestions for Authors
Dear Authors,
The manuscript is well-written and engaging, addressing an interesting topic. The Introduction offers a thorough overview of the subject, while the Discussion section is clearly and effectively presented with strong and fact-based conclusions. I have just a few comments, mainly regarding the Materials and Methods section.
Lines 64 to 66, sentence: „However, the influence of ultrasound on the structural and functional properties of cross-linking WPI by the dual oxidase system is limited. „ is unclear. Are you suggesting that there is no data available on the influence of ultrasound, or that its ultrasound impact is limited?
Lines 87 to 88, sentence: „The samples reached a final protein concentration of 5% (w/v).“ requires further clarification. Since the initial WPI concentration was 6%, did you dilute the sample to achieve the final 5% protein concentration? Could you explain the process in more detail?
What was the unit used to measure the free amino group content?
What equipment was used for FTIR spectroscopy analyses? For determining emulsifying properties, what equipment was used for mixing soybean oil and the sample?
Author Response
The manuscript is well-written and engaging, addressing an interesting topic. The Introduction offers a thorough overview of the subject, while the Discussion section is clearly and effectively presented with strong and fact-based conclusions. I have just a few comments, mainly regarding the Materials and Methods section.
Reply: Thanks for your positive comments.
Based on your valuable suggestions, we have done our best effort to revise and improve the quality of this manuscript. Thanks again!
Lines 64 to 66, sentence: „However, the influence of ultrasound on the structural and functional properties of cross-linking WPI by the dual oxidase system is limited. „ is unclear. Are you suggesting that there is no data available on the influence of ultrasound, or that its ultrasound impact is limited?
Reply: Thanks for your comments.
Research on the influence of ultrasound on the structural and functional properties of cross-linking WPI by the dual oxidase system is limited. We revised this sentence to make it clearer. Please refer to the revisions in lines 77-79 of the revised manuscript.
Lines 87 to 88, sentence: „The samples reached a final protein concentration of 5% (w/v).“ requires further clarification. Since the initial WPI concentration was 6%, did you dilute the sample to achieve the final 5% protein concentration? Could you explain the process in more detail?
Reply: Thank you for your suggestion.
WPI solution was first prepared at a concentration of 6%, then mixed with HRP, glucose oxidase and D-glucose at 200 U/g protein, 6 U/g protein and 0.05 mmol/g protein, respectively, and finally, deionized water was added to produce a final protein concentration of 50 g/L.
What was the unit used to measure the free amino group content?
Reply: Thanks for your comments.
The amount of free amino groups was expressed as nmol/mg protein, as indicated in Figure 1B.
What equipment was used for FTIR spectroscopy analyses? For determining emulsifying properties, what equipment was used for mixing soybean oil and the sample?
Reply: Thanks for your comments.
FTIR spectroscopy was performed using a Fourier transform infrared spectrometer (Nicolet iS50, Massachusetts, USA) and a tablet press. The emulsifying properties were determined by mixing the soybean oil with the sample using an IKA homogenizer (T18, IKA, Staufen, Germany). We added equipment to the methods section. Please see 2.7 and 2.12 in the revised manuscript.
Round 2
Reviewer 1 Report
Comments and Suggestions for Authors
Dear Authors, Thank you for your response. After reviewing the authors' revisions to my suggestions, I believe the manuscript is now ready for publication.
Regards